# Promoter Methylation of HIV Coreceptor-Related Genes CCR5 and CXCR4: Original Research

**DOI:** 10.3390/v17040465

**Published:** 2025-03-25

**Authors:** Anna Esman, Svetlana Salamaikina, Alina Kirichenko, Michael Vinokurov, Darya Fomina, Kirill Sikamov, Arina Syrkina, Anastasia Pokrovskaya, Vasily Akimkin

**Affiliations:** 1Federal Service for Surveillance on Consumer Rights Protection and Human Wellbeing, Central Research Institute of Epidemiology, 111123 Moscow, Russia; 2State Research Center—Burnazyan Federal Medical Biophysical Center of Federal Medical Biological Agency, 123098 Moscow, Russia; 3Medical Institute, Peoples’ Friendship University of Russia (RUDN University), 117198 Moscow, Russia

**Keywords:** epigenetic, HIV infections, CpG, methylation, HIV coreceptors, *CCR5*, *CXCR4*, ART, CD4+ T-cells

## Abstract

The persistence of human immunodeficiency virus (HIV) within viral reservoirs poses significant challenges to eradication efforts. Epigenetic alterations, including DNA methylation, are potential factors influencing the latency and persistence of HIV. This study details the development and application of techniques to assess CpG methylation in the promoter regions of the *CCR5* and *CXCR4* genes, which are key HIV-1 coreceptors. Using both Sanger sequencing and pyrosequencing methods, we examined 51 biological samples from 17 people living with HIV at three time points: baseline (week 0) and post-antiretroviral therapy (ART) at weeks 24 and 48. Our results revealed that *CXCR4* promoter CpG sites were largely unmethylated, while *CCR5* promoter CpGs exhibited significant variability in methylation levels. Specifically, *CCR5* CpG 1 showed a significant decrease in methylation from week 0 to week 48, while *CXCR4* CpG 3 displayed a significant decrease between week 0 and week 24. These differences were statistically significant when compared with non-HIV-infected controls. These findings demonstrate distinct methylation patterns between *CCR5* and *CXCR4* promoters in people living with HIV over time, suggesting that epigenetic modifications may play a role in regulating the persistence of HIV-1. Our techniques provide a reliable framework for assessing gene promoter methylation and could be applied in further research on the epigenetics of HIV.

## 1. Introduction

The lifelong persistence of human immunodeficiency virus (HIV) in people living with HIV (PLWH) is a global public health problem. Antiretroviral therapy (ART) significantly reduces mortality and the risk of transmission in PLWH through the suppression of viral replication [1,2,3,4], and HIV RNA counts are reduced in ART recipients with sustained virologic response to undetectable values (<50 copies/mL plasma). However, the provirus activates within a few weeks after the interruption of ART, leading to an increase in the viral load [5,6].

HIV-1 enters CD4+ T-cells via chemokine coreceptors and persists as a provirus. Recent data have indicated that resting CD4+ T-cells are a basic reservoir of the virus and a primary barrier to the elimination of HIV-1, even with ART [7]. Current therapies do not cure HIV infection, and the virus continues to persist in reservoirs.

The chemokine receptors CXCR4 and CCR5 serve as major coreceptors for the entry of HIV-1 into CD4+ T-cells. CD4+ T-cells can be categorized into distinct subsets, including naive, central memory, effector memory, and activated T-cells. Naive T-cells predominantly express CXCR4 [8], while memory and activated T-cells generally exhibit higher CCR5 expression, facilitating HIV replication [9].

HIV-1 primarily uses CCR5 as its coreceptor (R5-tropic HIV-1 strains), and only approximately 5% of all viral strains (X4-tropic HIV-1 strains) utilize CXCR4 as the primary coreceptor [10,11,12]. During the early stages of clinical HIV-1 infection, R5-tropic viruses are predominantly transmitted. In contrast, X4-tropic viruses are commonly detected in humans who have undergone multiple ART regimens and are in the later stages of infection, often with rapid disease progression [13,14,15,16,17].

The level of CCR5 expression on the surface of activated CD4+ T-cells [8,9,18] has been observed to be elevated in PLWH, when compared to healthy controls [17]. The high expression of the CCR5 gene significantly influences disease progression and hampers immune recovery during antiretroviral therapy [19]. Elevated CCR5 levels could activate resting T-cells, triggering apoptosis and their subsequent destruction. On the other hand, therapies targeting CCR5 expression could slow down the depletion of the CD4+ T-cell pool; however, they may not exhibit direct antiviral effects [20].

To the contrary, the reduced expression of CCR5 correlates with decreased HIV-1 concentrations within cells [21]. Moreover, the deletion of 32 base pairs in the *CCR5* gene resulted in a truncated protein that is unable to function properly, thereby blocking the entry of R5-tropic HIV variants and providing resistance to approximately 50% of HIV-1 strains [22].

Epigenetic modifications, including DNA methylation and histone modification, play a crucial role in the regulation of gene expression without altering the underlying DNA sequence. These modifications influence various cellular processes, such as development, differentiation, and responses to environmental stimuli. The main forms of epigenetic inheritance include genomic imprinting, DNA methylation at CpG loci, and histone modifications [23].

DNA methylation, particularly cytosine methylation (5 mC) at CpG loci within gene promoter regions, is strongly associated with transcriptional silencing. This process is a key mechanism of epigenetic regulation and can prevent the binding of transcription factors to DNA, thus reducing the expression of genes [24,25,26,27,28,29]. Methylation near the transcription initiation site is especially impactful in regulating gene activity as it disrupts the normal functioning of regulatory elements within these regions.

In the context of HIV-1, gene methylation plays a pivotal role in regulating coreceptor expression on the cell surface. The methylation of genes that encode HIV coreceptors can reduce their expression, potentially preventing HIV-1 from entering the cell. These epigenetic changes do not involve alterations in the DNA sequence but can significantly influence cellular functions and viral susceptibility. Moreover, HIV-1 latency and reactivation may be regulated by changes in the chromatin structure near the viral promoter located at the 5′ long terminal repeat (LTR), further highlighting the importance of epigenetic regulation in the viral life cycle [30].

In this way, increased DNA methylation in cis-regulatory modules of the *CCR5* gene reduces its activity. A significant correlation exists between the methylation level of the *CCR5* gene and its expression on the surface of T-cells [20], with lower expression levels being associated with the demethylation of these cis-regulatory modules. Naive T-cells—which are activated via HLA receptors in response to infection—show increased *CCR5* expression that directly supports T-cell activation. While hypermethylation of the gene is associated with lower viral loads and higher CD4+ T-cell counts, the cis-regulatory modules in PLWH remain demethylated when compared to those with normal CD4+ T-cell counts [31]. The demethylation of the *CCR5* gene cis-modules results in increased gene expression, which raises the amount of CCR5 protein on the cell membrane, thus facilitating the entry of HIV-1 into uninfected CD4+ T-cells [32]. In this way, the human immune system inadvertently supports HIV-1 replication. However, ART—which suppresses viral replication—has been linked to an increased methylation level of the *CCR5* gene [20].

Although studies on the impacts of *CXCR4* gene methylation on HIV-1 replication are limited, some data suggest a strong correlation between *CXCR4* expression levels and the potential for entry of HIV-1 into cells [33]. The activation of CD4+ T-cells can reduce the number of CXCR4 receptors on the cell surface, thus limiting the spread of the infection [11]. Moreover, decreased *CXCR4* gene expression is associated with progression of the disease [17].

In this study, we develop methodologies and assess the impacts of cytosine methylation at CpG loci within *CCR5* and *CXCR4* genes promoter regions in ART-treated HIV-positive individuals compared to healthy, HIV-negative controls.

## 2. Materials and Methods

### 2.1. Study Population

Depersonalized samples from 17 antiretroviral-naïve PLWH collected during 2021–2023 and 27 non-HIV-infected individuals during 2024 at the infectious disease clinics of the Central Research Institute of Epidemiology (CRIE) were used in this study. HIV was diagnosed in accordance with country-specific measures based on the WHO guidelines, in particular, based on two positive ELISA tests confirmed via immunoblot [34,35].

Inclusion criteria were as follows: age over 18 years, viral load <500,000 copies/mL, CD4+ T-cell count >200 cells/mm^3^, less than 3 visits, written informed consent.

The first-line ART regimen prescribed was lamivudine/dolutegravir (3TC/DTG). Whole-blood samples from PLWH were obtained at baseline (week 0) and after ART started (weeks 24 and 48). In summary, this study included 51 samples from 17 HIV-1 PLWH collected during three visits.

### 2.2. Biological Samples

CD4 T-cell counts were obtained using a FACSCalibur flow cytometer (Becton Dickinson, Franklin Lakes, NJ, USA), according to the standard manufacturer’s protocols.

Whole human blood samples were examined for the presence of HIV DNA using the AmpliSens^®^ “DNA-HIV-FL” reagent kit (Amplisens, CRIE, Moscow, Russia), which includes a plasmid calibrator with a known concentration of human *β-globin* gene and HIV *pol* gene.

Samples purified with hemolytic reagents (Amplisens, CRIE, Moscow, Russia) were used for selective lysis of blood erythrocytes during pre-processing of clinical materials of whole peripheral blood prior to DNA extraction.

DNA extraction was carried out using the “RIBO-prep” kit (Amplisens, CRIE, Moscow, Russia), according to the manufacturer’s instructions.

The concentration of the starting DNA was measured via real-time PCR for the *β-globin* gene using plasmid calibrators with concentrations of 10,000 and 100 copies per mL. The obtained values were then converted from “copies/µL” to “ng” using the SciencePrimer tool’s copy number calculator for real-time PCR [36] (where 1 ng DNA is equivalent to approximately 300 copies). The average concentration of starting DNA before the bisulfite conversion reaction was 9000 copies per μL (or 30 ng).

Bisulfite conversion was performed on samples using the “EpiTect Bisulfite KIT” (QIAGEN, Hilden, Germany). DNA samples were kept at —20 °C after bisulfite conversion and eluted 10× in the TE buffer (AmpliSens, CRIE, Moscow, Russia) for PCR analysis.

It is necessary to note that incomplete bisulfite conversion of DNA results in the interpretation of unconverted cytosines as being methylated in assays [37]. Large amounts of DNA (>10 μg/mL) could decrease the conversion rate by depleting the available bisulfite in the reaction mix and increasing the pH due to the formation of NH4+ as a by-product of the reaction.

### 2.3. CpG Loci Selection

The database of the University of California, Santa Cruz (UCSC), was used to select the *CCR5* and *CXCR4* promoter regions and related predicted transcription factors (JASPAR CORE) [38]. The whole *CXCR4* promoter region was chosen for this study. According to the database, the *CCR5* gene contains 2 promoter regions: promoter region 1 follows the first exon and produces a truncated product that is primarily expressed in naïve T-cells and in memory T-cells [39], while promoter region 2 contributes to transcription of a product with exon 1, which is typical for T-cells. We studied promoter region 2 as the results of the study conducted by Gornalusse et al. showed that hypermethylation of this DNA region correlates significantly with the *CCR5* expression level [20].

Target loci were chosen by comparing promoter sequences from the EPD (Eukaryotic Promoter Database) [40], which contains experimentally confirmed regions at the beginning of transcription, including sequences from CpG islands; that is, regions where CpG is present more frequently than in other regions.

Coordinate data and CpG identifiers were obtained from the Illumina GRCh37.p13 genome assembly methylation array (GEO Access Viewer) [41] and NCBI database [42].

A typical procedure for methylation research would include a gene expression study, the identification of genes that are downregulated, and an NCBI search (National Center for Biotechnology Information) [42] to locate the promoter sequence and transcription start site of the gene. The selected genomic sequence is then used to aid in the selection of primers.

Detailed descriptions of the chosen loci and predicted transcription factors are presented in Table 1 and Table 2 in which the CpGs assessed in our study are indicated by blue color.

### 2.4. Primer Design

Selected CpG loci were detected using pyrosequencing and Sanger sequencing methods. Figure 1 shows the common scheme of the oligonucleotide positions.

NB! CpGs at annealing sites should be avoided in oligonucleotide design. The length of the amplicon for pyrosequencing was in the range of 100–250 bp [43].

**Figure 1 viruses-17-00465-f001:**
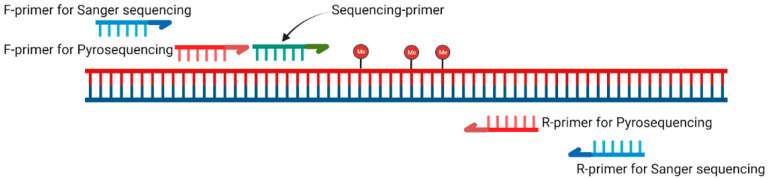
Common scheme of techniques for determining the methylation level of CpG loci (created with BioRender [44]).

Oligonucleotides were designed according to standard conditions. The Tm mean calculated using the OligoAnalyzer™ Tool [45] was 60 °C. The CG content was rather low due to the features of the converted DNA product, with an average of 33%. Table 3 summarizes the sequences of the oligonucleotides used in this study.

### 2.5. PCR Conditions

The “Tercyc” amplification system (DNA-technology, Protvino, Russia) was used for PCR sample preparation before sequencing. The amplification program was as follows: hold 95 °C for 15 min, then 45 cycles of 95 °C for 5 s, 60 °C for 20 s, and 72 °C for 5 s.

PCR reactions were carried out in a final volume of 25 μL, containing 5 μL of PCR-mix-1 (dNTPs and primers; the concentration of primers was experimentally determined as 0.7 µM), 10 µL of PCR-mix-2 (Tris-buffer and MgCl_2_), and 0.5 µL of TaqF polymerase. All used reagents and kits were developed in the Central Research Institute of Epidemiology, Federal Service for Surveillance on Consumer Rights Protection and Human Wellbeing Russian Federation (AmpliSens, Moscow, Russia). Finally, 10 μL of bisulfite-treated DNA sample was added to the prepared tubes.

Completely methylated DNA and fully unmethylated DNA (as negative control) were used as controls for amplification [46,47] (QIAGEN, Hilden, Germany).

### 2.6. Pyrosequencing Conditions

Sample preparation was carried out using the PyroMark Q24 vacuum station (QIAGEN, Hilden, Germany) and a “Amplisens^®^ Pyroscreen” sample preparation kit (Amplisens, CRIE, Moscow, Russia), according to the manufacturers’ recommendations.

The reaction was performed with 5 μL of the PCR products. Incubation with the sequencing primer was carried out at 80 °C for 2 min (at a concentration of 7.5 pmol/µL).

Pyrosequencing was performed using the “PyroMark Gold Q24 Reagents kit” (Qiagen, Hilden, Germany).

Methylation levels of CpGs via pyrosequencing were calculated automatically using the PyroMark Q24 software (Qiagen, Hilden, Germany). Quantitative bisulfite pyrosequencing determines DNA methylation levels through analyzing artificial “C/T” SNPs at CpG sites within a specific pyrosequencing assay [43].

### 2.7. Sanger Sequencing Conditions

The amplification system “T100” (BioRad Laboratories, Hercules, CA, USA) was used for PCR sample preparation with the following steps: 96 °C for 1 min, followed by 25 cycles of 96 °C for 10 s, 50 °C for 5 s, 60 °C for 4 min. Sanger sequencing was performed using the “AmpliSense^®^ HIV-Resist-Seq” reagent kit (Amplisens^®^, CRIE, Moscow, Russia) and an Applied Biosystems 3500 genetic analyzer (LifeTechnologies, Waltham, MA, USA).

### 2.8. Tropism Assay

The AmpliSens^®^ “HIV-Resist-Seq” reagent kit (AmpliSens, Moscow, Russia) was used for DNA extraction from whole-blood samples, as well as for amplification and sequencing of the third variable region (V3) of the HIV-1 envelope protein gp120 (6960–7370 bp according to the HXB-2 strain; GenBank accession number K03455).

Viral tropism was determined using the algorithm of the Geno2pheno genotypic tool [48], with 20% false positive rate (FPR).

### 2.9. Statistical Methods

Methylation analysis of CpGs via Sanger sequencing was performed according to the protocol of Parrish et al. [49]. Methylation level data for each CpG locus were preliminarily processed in Microsoft Excel [50]. All statistical tests were run using R 4.2.2 [51,52]. Descriptive statistical analysis, two-way ANOVA with Dunnett test (for comparison of gene methylation levels between control non-HIV-infected individuals and PLWH at baseline/week 0, week 24, and week 48), and Mann–Whitney U-test (for comparison of CpGs methylation level in PLWH at baseline/week 0, week 24, and week 48 with each other) were employed for evaluation of the data. The results were considered statistically significant at *p* < 0.01. Data visualization was performed using the graph application provided at BioRender.com [44].

## 3. Results

### 3.1. Sample Description

First, we analyzed the viral load, viral reservoir size, and CD4+ T-cell counts in samples collected from PLWH at baseline (week 0) and after ART (weeks 24 and 48).

Viral load at the first visit for PLWH varied from 1371 to 208,538 copies/mL, with a median of 38,181 copies/mL. All PLWH presented viral suppression (HIV plasma RNA < 50 copies/mL) at weeks 24 and 48. There were no treatment-related adverse effects requiring regimen changes, and there was no clinical progression of infection in the sense of HIV-associated diseases. All PLWH reached the virological efficacy of ART by week 24 of treatment and remained at week 48.

HIV-1 viral reservoir volume decreased systematically over the course; however, there was no statistically significant reduction.

Normally, CD4+ T-cell counts in non-HIV-infected individuals range from 500 to 1500 cells per mm^3^. There was a positive trend of CD4+ T-cell counts (Figure 2).

Information about viral load, size of the viral reservoir (measured in copies of HIV DNA per million cells), and CD4+ T-cell count (cells/mm^3^) for all PLWH has been collected in [50].

### 3.2. Development of CpG Locus Methylation Techniques

Pyrosequencing and Sanger sequencing methods were used to assess CpG locus methylation in promoter regions of the *CCR5* and *CXCR4* genes. Pyrosequencing is undoubtedly more accurate in measuring the C/(C + T) ratio of peak heights, and this method is considered the “gold standard” for quantitative allele quantification at single base resolution [53].

Figure 3 presents an example of the *CXCR4* gene promoter sequence. The pyrogram obtained from the PyroMarkQ24 (Qiagen, Hilden, Germany) is shown in the upper part of the figure, the same region obtained via Sanger sequencing is in the middle, and the expected sequence is shown in the lower part of the figure. Lines link the related CpG loci on the pyrogram and chromatogram.

Considering the CpG locations in the *CCR5* gene’s second promoter region, several combinations of oligonucleotides were used to read 100–250 bp amplicon length sites via pyrosequencing. These combinations are presented in Section 2.4 (Methodology Design/Primers Design).

Fully methylated control reactions containing bisulfite-treated DNA (QIAGEN, Hilden, Germany) were on average 88% and 93.5%, while fully unmethylated control reactions with bisulfite-treated DNA (QIAGEN, Hilden, Germany) were on average of 1% and 4.8% in assays for *CCR5* and *CXCR4,* respectively. Comparison of the results obtained using the pyrosequencing and Sanger sequencing methods indicated no statistically significant differences.

### 3.3. Application Results of the Developed Techniques

The results obtained using the developed pyrosequencing and Sanger sequencing techniques did not reveal any significant differences. Therefore, for the purposes of this study, we chose to use data generated only using the Sanger method.

The mean methylation levels of the *CCR5* and *CXCR4* promoters in PLWH at baseline (week 0), week 24, and week 48 differed with respect to those of non-HIV-infected individuals in a statistically significant manner (*p* < 0.0001; Figure 4). We also observed that methylation levels of *CXCR4* promoter regions were significantly lower than those for *CCR5*.

Next, we analyzed all CpGs in the *CCR5* gene separately in PLWH at baseline (week 0), week 24, and week 48. The CpG 1 methylation level of the CCR5 promoter ranged from 33% to 99%, that for CpG 2 from 4% to 83%, that for CpG 3 from 0 to 42%, and that for CpG 4 from 1% to 21%. Analysis of the methylation of CpGs in the promoters of *CCR5* revealed statistically significant differences in methylation levels between PLWH at different follow-up stages, specifically between the week 0 and week 48 groups, at CpG 1 (*p* = 0.0015; Figure 5).

Analysis of the *CXCR4* promoter region indicated that the CpG 1 methylation level ranged from 0% to 17%, that for CpG 2 from 0% to 21%, that for CpG 3 from 0% to 21%, that for CpG 4 from 0% to 25%, that for CpG 5 from 1% to 17%, that for CpG 6 from 1% to 16%, that for CpG 7 from 0% to 11%, and that for CpG 8 from 0% to 15%. Statistical differences were observed in CpG3 between baseline (week 0) and week 24 (*p* = 0.002; Figure 5).

Based on these results, we compared the methylation levels of *CCR5* (CpG 1) and *CXCR4* (CpG 3) CpGs between baseline (week 0) and week 48, as well as between week 0 and week 24, according to the statistical differences with respect to non-HIV-infected individuals. Statistical differences were observed for the following groups via two-way ANOVA with Bonferroni multiple comparisons test (Table 4).

We compared the methylation levels in promoter regions of the *CCR5* and *CXCR4* genes in our group of patients split by tropism type. A regular two-way ANOVA was performed. There was a significant difference in methylation level of the *CXCR4* promoter region between patients with CXCR4 tropism type at week 24 and patients with CCR5 tropism type at week 0 (Figure 6).

## 4. Discussion

Genome DNA methylation is not uniformly distributed: both promoter and CpG islands are typically hypomethylated, whereas the extent of methylation in gene bodies is higher than that in intergenic regions. While early studies suggested that DNA methylation represses gene expression, a growing body of evidence has indicated that DNA methylation plays a dual role—both inhibitory and permissive—depending on the genomic region in which DNA methylation occurs [54].

The association between viral replication and *CXCR4* and *CCR5* expression levels suggests a possible direct effect of methylation of these genes on the progression of HIV-1 within human immune cells [17]. The activities of the *CCR5* and *CXCR4* receptor genes largely regulate the entry of HIV-1 into cells. The gene methylation of the promoter regions decreases the concentration of unintegrated HIV-1 in cells and could contribute to viral eradication [19], and the hypermethylation of *CCR5* and *CXCR4* promoters can reduce both the concentration of HIV-1 RNA inside cells and the risk of infecting resting CD4+ T-cells.

Methylation levels can be analyzed using various methods, such as pyrosequencing, which uses bisulfite-treated DNA. However, the Sanger method is more convenient when the required multiple targets are located more than 60–70 nucleotides apart as a well-read fragment for pyrosequencing is usually limited to about 50 nucleotides. We presented combinations of primers for pyrosequencing and Sanger sequencing methods in this study. In the case of Sanger sequencing, it will be sufficient to use either one or two pairs of primers to suppress the “noise” that arises during result generation.

We also compared these methods on designed primers for the detection of methylation levels in *CCR5* and *CXCR4* promoter regions. The results indicated no differences between the used methods for detection of CpG methylation. While some research groups prefer pyrosequencing as the optimal method, Sanger sequencing also yields satisfactory results [55].

In this study, we noticed differences in the methylation level between the observed CpGs in the *CCR5* promoter region and hypomethylation of the *CXCR4* promoter region. These findings indicate an interesting area for further research. When DNA methylation affects the promoter region, the transcription of the gene is usually suppressed, whereas actively expressed genes tend to have unmethylated promoters. In our data, we found that specific CpGs were more accessible to transcription factors, as identified via JASPAR CORE data [42], where these loci correspond to binding motifs for the transcription factors listed in Table 1 and Table 2. This suggests that differential methylation at these loci may influence gene expression through modulating the recruitment of these specific transcription factors.

For example, transcription factors predicted to bind in the *CCR5* promoter region—such as REL, a member of the NF-κB/Rel family—play a critical role in the maintenance of HIV-1 latency, suggesting that methylation of these binding sites could influence the ability of HIV-1 to persist in a latent state [56]. Similarly, FOXD3 acts as a negative regulator of HIV-1 replication in mononuclear phagocytes through suppressing genes involved in viral replication, such as OTK18. The upregulation of FOXD3 during HIV infection could explain the restricted replication of the virus in certain immune cells like microglia [57].

Furthermore, Tfcp2l1 is a member of the Grainyhead-like transcription factor family, which regulates both cell cycle progression and the expression of HIV-1 genes [58]. Its binding motifs in the promoter regions of *CXCR4* could play a role in controlling the expression of these coreceptors, thus impacting the entry of HIV-1 into host cells. Additionally, Nrf1 plays a crucial role in regulating *CXCR4* promoter activity, and its involvement could lead to enhanced *CXCR4* expression during immune activation, potentially contributing to increased HIV-1 replication and disease progression [59].

Although we observed some statistically significant differences in methylation levels between PLWH and non-HIV-infected controls, these results could be due to random variations, considering the small sample size in our study and other factors that may influence methylation levels, such as age, the status of other genes, and so on. Although we observed differences in methylation levels between the *CCR5* and *CXCR4* promoter regions, we could not clearly conclude whether these results are associated with a protective effect. We can only hypothesize that the activity of the *CCR5* gene was reduced in the analyzed samples.

For confirmation of the obtained results, further investigation of the expression levels of these genes is necessary.

Epigenetic modification research focused on the chemokine receptor group could reveal the connections between HIV tropism and the methylation levels of CpGs in promoters, which could further uncover new pathways for the elimination of HIV. The eradication of HIV-1 in PLWH will be impossible without solving the problem of the persistence of the virus in its reservoirs.

Studying the differences in DNA methylation patterns between non-HIV-infected individuals and PLWH, as well as replicating the study’s findings and discovering new host gene methylation targets associated with HIV tropism and the response of the immune system to HIV infection, will not only allow for the development of new therapeutic approaches to determine effective ART tactics in naive patients but also the expansion of the fundamental understanding regarding the mechanisms underlying the host’s regulation of HIV infection.

## Figures and Tables

**Figure 2 viruses-17-00465-f002:**
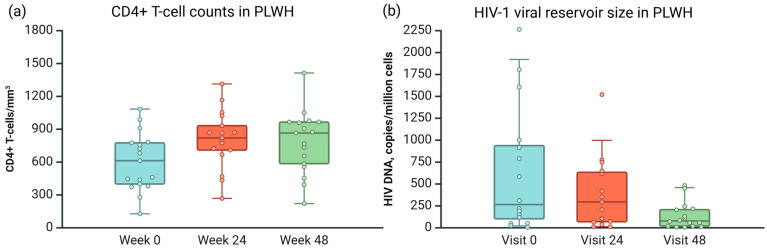
(**a**) CD4+ T-cell counts per mm^3^ and (**b**) HIV-1 viral reservoir size in samples collected from PLWH at baseline (week 0) and after ART (weeks 24 and 48). Created in BioRender.com [44].

**Figure 3 viruses-17-00465-f003:**
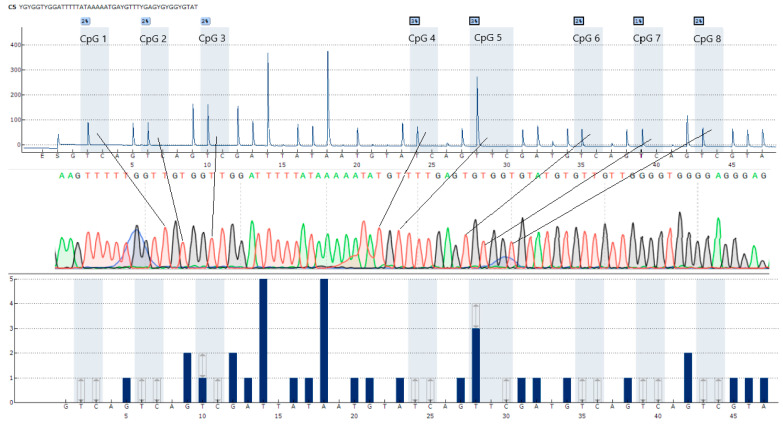
An example *CXCR4* promoter region sequence.

**Figure 4 viruses-17-00465-f004:**
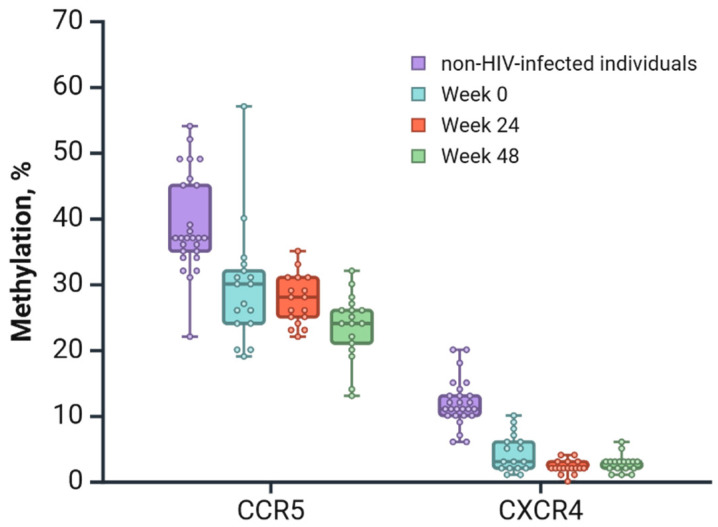
The average methylation level in *CCR5* and *CXCR4* promoter regions in samples collected from PLWH, assessed at baseline (week 0), week 24, and week 48, compared to non-HIV-infected individuals. Created in BioRender.com [44].

**Figure 5 viruses-17-00465-f005:**
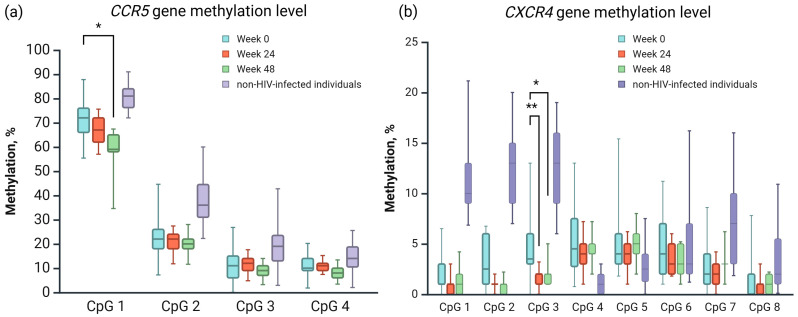
Methylation percentage for each separate CpG locus in (**a**) *CCR5* and (**b**) *CXCR4* promoter regions compared between the groups of this study: samples collected from non-HIV-infected individuals and PLWH, assessed at baseline (week 0), week 24, and week 48. * Methylation level comparison between PLWH at baseline (week 0) and week 48. ** Methylation level comparison between PLWH at baseline (week 0) and week 24. Created in BioRender.com [44].

**Figure 6 viruses-17-00465-f006:**
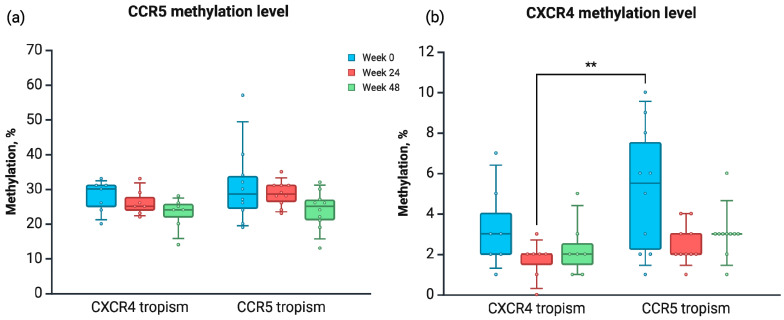
The average methylation level in (**a**) CCR5 and (**b**) CXCR4 promoter regions in samples collected from PLWH, assessed at baseline (week 0), week 24, and week 48, split by tropism type (HIV sequence analysis). Created in BioRender.com [44]. ** *p* < 0.01.

**Table 1 viruses-17-00465-t001:** CpG coordinates and predicted transcription factors located in the *CCR5* gene.

*CCR5* (GRCh37.p13) Chr 3—NC_000003.12					
cg22984586 *	CpG 1	CpG 2	CpG 3	CpG 4	cg07616471 *
46411541	46412248	46412333	46412454	46412478	46413711
**JASPAR CORE-predicted transcription factors**					
	REL, FOXD3, FOXF2	–	–	ZNF784	

* CpG identifiers according to the GEO Access Viewer.

**Table 2 viruses-17-00465-t002:** CpG coordinates and predicted transcription factors located in the *CXCR4* gene.

*CXCR4* (GRCh37.p13) Chr 2—NC_000002.12									
cg25982140 *	CpG 1	CpG 2	CpG 3	CpG 4	CpG 5	CpG 6	CpG 7	CpG 8	cg12311057 *
136875681	136875732	136875734	136875738	136875756	136875761	136875765	136875767	136875770	136875897
**JASPAR CORE-predicted transcription factors**									
	ERF::NHLH1, Tfcp2l1	ERF::NHLH1, Tfcp2l1	ERF::NHLH1, Tfcp2l1	FOXE1	—	—	Nrf1	Nrf1	

* CpG identifiers according to the GEO Access Viewer.

**Table 3 viruses-17-00465-t003:** Oligonucleotides used in this study.

Gene	Primers	P/S	5′-Sequence-3′	Amplicon Size	Dispensation Order for Pyrosequencing
*CCR5*	CCR5-Fp	P	TATGATTGATTTGTATAGTTTATTTGGTTA	Fp + Rp 121	–
	CCR5-Rp	P	biotin-CTCATCTCAAAAACTAACTAAC		
	CCR5-F1p	P	GTAGTGGGATGAGTAGAGAATA	F1p + Rp 331	–
	CCR5-S2	P	GAAGAATTGTTTTTTGATTTTTTT	–	YGTTTTTAATATAT
	CCR5-S3	P	GGTTAGAAGAGTTGAGATATT	–	YGTTTTTTTATAAGAAATTTTTTTYGGTAAG
	CCR5-Fs	S	GTGGGTTTTTGATTAGATGAATGTA	Fs + Rs 517	–
	CCR5-Rs	S	CCAAACTATAACCCTTTCCTTATCTT		
*CXCR4*	CXCR4-Fp	P	GTG GTT ATT GGA GTA TTT AGG T	Fp + Rp 168	–
	CXCR4-Rp	P	biotin-TCT ACC CCT CTC CCC CA		
	CXCR4-S	P	GTTAATAAATTGAAGTTTTTGGT	–	YGYGGTYGGATTTTTATAAAAATAYGTTTYGAGYGYGGYGTAT
	CXCR4-Fs	S	ATGATAAAGTAGGTTGAAATTGGATTT	Fs + Rs 376	–
	CXCR4-Rs	S	TCCCTCAAACTTAAAAAATACCTCTA		

P/S—pyrosequencing/Sanger; F—forward primer; R—reverse primer; S—sequencing primer; Y—C/T.

**Table 4 viruses-17-00465-t004:** Methylation level differences in PLWH with respect to baseline values.

	Week 0	Week 24	Week 48	*p*-Val	CI 95%
CCR5 (CpG 1)	71.24	66.88	58.24	0.0059 *	2.3473–23.6527
CXCR4 (CpG 3)	4.75	1.53	2.12	0.0013 **	0.8614–5.5798
				0.0169 *	0.2731–4.9916

* Methylation level comparison between PLWH at baseline (week 0) and week 48. ** Methylation level comparison between PLWH at baseline (week 0) and week 24.

## Data Availability

The original data presented in the study are openly available in FigShare at https://figshare.com/s/2fce38ebe705a8501770 (accessed on 1 March 2025).

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
