# Peer review of "Promoter Methylation of HIV Coreceptor-Related Genes CCR5 and CXCR4: Original Research"

_viruses, 2025, doi:10.3390/v17040465_

Round 1
Reviewer 1 Report (Previous Reviewer 1)
Comments and Suggestions for Authors
The Introduction needs to be thoroughly revised. It still contains several significant inaccuracies.The authors must revise the subclasses of CD4 lymphocytes (e.g., line 56: activated cells are NOT equivalent to memory cells). They should reassess which lymphocytes predominantly express the CXCR4 or CCR5 coreceptors. (e.g., in the revision, lines 57-58—containing a correct statement—were deleted, while lines 60–61, which provide misleading information, were retained). The description of tropism of HIV-tropism must be improved, specifying which strains are most prevalent and which coreceptor they utilize (e.g., lines 46–47: the most common strains do NOT use CXCR4). These examples have been provided to facilitate a comprehensive and consistent overall revision.
In the Materials and Methods section, redundant details should be removed when already included in the protocols provided with the commercial kits. In such cases, the phrase "according to the manufacturer’s instructions" should be retained. Do not remove this phrase, and keep all necessary details.
Figure 5 is unclear, as is its explanation (lines 344–349).
The Discussion must be revised to reflect the results and their potential implications for pathogenesis and treatment.
It is recommended to modify how patients with HIV are dedined . The International AIDS Society (IAS) advocates for people-first language to avoid stigmatizing terminology. The term "people living with HIV" (PLWH) should be used instead.
Comments on the Quality of English LanguageThe english should be improved to more cleraly express the research
Author Response
The Introduction needs to be thoroughly revised. It still contains several significant inaccuracies.The authors must revise the subclasses of CD4 lymphocytes (e.g., line 56: activated cells are NOT equivalent to memory cells). They should reassess which lymphocytes predominantly express the CXCR4 or CCR5 coreceptors. (e.g., in the revision, lines 57-58—containing a correct statement—were deleted, while lines 60–61, which provide misleading information, were retained). The description of tropism of HIV-tropism must be improved, specifying which strains are most prevalent and which coreceptor they utilize (e.g., lines 46–47: the most common strains do NOT use CXCR4). These examples have been provided to facilitate a comprehensive and consistent overall revision.
- Thank you for your comments, we have made significant changes.
In the Materials and Methods section, redundant details should be removed when already included in the protocols provided with the commercial kits. In such cases, the phrase "according to the manufacturer’s instructions" should be retained. Do not remove this phrase, and keep all necessary details.
- We kept the phrase and deleted unnecessary details
Figure 5 is unclear, as is its explanation (lines 344–349).
- We have corrected the inscription
The Discussion must be revised to reflect the results and their potential implications for pathogenesis and treatment.
- We have added information
It is recommended to modify how patients with HIV are dedined . The International AIDS Society (IAS) advocates for people-first language to avoid stigmatizing terminology. The term "people living with HIV" (PLWH) should be used instead.
- Thanks, we fixed this throughout the text
Reviewer 2 Report (Previous Reviewer 4)
Comments and Suggestions for Authors
The authors have addressed all my concerns raised to my satisfaction. In addition, all revisions made have improved the manuscript's clarity. I recommend the publication of the manuscript.
Author Response
Thank you so much for your comments, which made the manuscript better
Round 2
Reviewer 1 Report (Previous Reviewer 1)
Comments and Suggestions for Authors
The revision has slightly improved the manuscript, but further modifications are needed.
COMMENTS
- The references must be appropriately mentioned (see, for example, references 10 and 22).
- The revision (essentially the last sentence of the discussion) introduces irrelevant information unrelated to the study and does not adequately address the request to discuss the potential therapeutic implications of the study's findings.
The revision has slightly improved the manuscript, but further modifications are needed.
COMMENTS
- The references must be appropriately mentioned (see, for example, references 10 and 22).
- The revision (essentially the last sentence of the discussion) introduces irrelevant information unrelated to the study and does not adequately address the request to discuss the potential therapeutic implications of the study's findings.
Author Response
The references must be appropriately mentioned (see, for example, references 10 and 22).
- Thank you, we fixed this flaw.
The revision (essentially the last sentence of the discussion) introduces irrelevant information unrelated to the study and does not adequately address the request to discuss the potential therapeutic implications of the study's findings.
-We have removed unnecessary information. We have made changes in the review format

This manuscript is a resubmission of an earlier submission. The following is a list of the peer review reports and author responses from that submission.
Round 1
Reviewer 1 Report
Comments and Suggestions for Authors
The study by Esman and colleagues analyzes the methylation status of two coreceptors, CXCR4 and CCR5, which are crucial for HIV entry into target cells. Resolving some of the questions raised by this study could be instrumental in developing strategic therapies for HIV treatment.
Comments
Introduction: The introduction should be revised to eliminate repetition and update references.
Results:
-Lines 264-265: The text does not match the observations in Figure 1. It needs to be revised.
-Lines 309-315: The text does not correspond with the data presented in Figures 4 and 5 regarding the reported methylation percentages. Additionally, statistical significance should be included in the figures.
-Figures 4 and 5: Of particular relevance is that both Figure 4 (overall percentages of methylation for the promoter of CCR5 coreceptor) and Figure 5 (percentage of methylation at individual sites analyzed) show a decrease in methylation of the CCR5 promoter during ART. However, in the abstract, the authors state the opposite (i.e. that the promoter shows a significant increase in methylation levels). The authors should revise the text accordingly and adjust the discussion.
-Lines 330-333: The statement here should be supported by data (through either a figure or an appropriate table).
Discussion:The discussion focuses on the significance of methylation. It should instead be more oriented towards the study results and the implications of CpG site methylation.
To strengthen the study's outcomes and validate its relevance, the authors should relate the methylation levels at the observed sites with the expression levels of the two coreceptors in peripheral blood cells, the tropism of the infecting virus (R5- or X4-tropic), and the levels of intracellular viral mRNAs in the cohort of people living with HIV they studied
Reviewer 2 Report
Comments and Suggestions for Authors
In this paper, Esman et al. present evidence that promoter regions for CXCR4 and CCR5 genes (encoding coreceptors for HIV-1 entry) have different levels of methylation in HIV-infected vs uninfected individuals. Additionally, methylation levels in these promoters changes over time with ART treatment in HIV-1 positive individuals. The authors suggest that methylation level could affect levels of gene products, which in turn could alter HIV-1 infection. Although the paper shows an interesting trend, the paper would benefit from further investigation and also lacks a clear message backed up by data. More detailed opinions are listed below:
The introduction is a bit confusing and repetitive to read. For example lines 48-49 and 52-53 seem to say the same thing.
Some lines seem to contradict each other – for example lines 56-57 states that naïve T cells have higher CXCR4 expression whilst lines 59-60 then say that naïve cells have reduced CXCR4 expression.
Some of the introduction seems to be unrelated to the rest of the contents of the paper, for example lines 82-83 about HIV latency.
Line 97 states that the ‘human immune system promotes the replication of HIV-1’ which is very misleading.
The materials and methods contain a lot of extra details about manufacturer’s protocols that could be removed.
It is unclear where the conclusion that ‘1ng DNA is approximately 300 copies’ (Line 148) came from. This seems like a very large underestimate. The number of copies in 1ng would depend on the length of the DNA, and for example for the beta globin gene would be approximately 6x10^8 copies, not 300.
“HIV-1 individuals” should be corrected to “HIV-1 positive individuals” (e.g. Line 256).
Some data in the results described in the text should also be shown in a figure – e.g. Lines 256, 263.
It should be stated if the example in Figure 3 is from the cohort. Labels could be added to the Figure in Figure 3 to make it more clear.
In Figure 4, it should be made clear that the control group is uninfected individuals.
Line 315 – might be useful to add significance stars to the graph.
Figure 6 – Why are the data points all whole numbers?
Line 330-333 are very unclear and seem to have data missing.
One large assumption of the paper is that methylation levels could affect levels of the gene products (CXCR4 or CCR5 protein) being expressed. However, there is no attempt to measure levels of these proteins in samples, e.g. by flow cytometry and correlate this to methylation levels.
Comments on the Quality of English Language
The English is largely accurate, but the phrasing is at times complicated and unclear and could be improved to make the paper more readable.
Reviewer 3 Report
Comments and Suggestions for Authors
Cytosine methylation in promoter regions of genes is an important epigenetic control in response to virus infection. This brief study shows a technology advance to study the methylation status in the promoter regions of CCR5 and CxCR4 genes as it relates to HIV-1 latency and reactivation. Fig 1 is a schematic for studying methylation levels. Figure 2 is the CD4+ counts and Figure 3 is example of the promoter region. These figures could be combined into a single figure with panels A &B &C. While Figure 2 suggests a positive trend between 0 and 24 wks which stabilizes at 48 wks, the statistical analysis is not presented. Figures 4 and 5, could also be combined into panels A and B. These demonstrate changes in methylation of 2 gene promoters.
The discussion speculates about a number of examples where gene regulation could influence HIV expression and I think it would be valuable to include some of these experiments in this study
Reviewer 4 Report
Comments and Suggestions for Authors
In this study, Esman et al. developed a technique to assess the methylation status in the promoter regions of HIV-1 co-receptor genes CCR5 and CXCR4 in individuals undergoing ART, comparing these findings to healthy, HIV-negative controls. The expression of CCR5 and CXCR4 is known to correlate with HIV-1 infection and its pathogenesis. Specifically, CCR5 expression is regulated by methylation within its gene, which may influence HIV-1 infection susceptibility. Therefore, a reliable method for evaluating methylation levels in CCR5 and potentially CXCR4 genes in HIV-1 patients could be important for estimating the susceptibility of CD4+ T cells to HIV-1. They found that pyrosequencing and Sanger sequencing with their custom-designed primers yield comparable results for methylation analysis in these genes. Additionally, Sanger sequencing revealed that CCR5 gene methylation is more pronounced than CXCR4 gene methylation, with significant enrichment at CpG 1, a CpG island of CCR5 gene, in HIV-1 patients. However, this study lacks appropriate controls in some figures, despite the manuscript mentioned differences between sample and control groups. Furthermore, although the authors sought to develop a new technique, they did not compare their primers with those designed previously by another group (Gornalusse et al.). Additionally, a critical discrepancy remains unaddressed: while Gornalusse et al. reported that ART increases CCR5 gene methylation in HIV-1 patients, this study instead shows a decrease in methylation with ART treatment. The manuscript also lacks a clear description about what questions did the authors address and a clear conclusion, and some references cited are not appropriate. Therefore, there are a number of things that should be addressed in the manuscript to be published from Viruses.
Major comments
1. Figures: The authors should add the figure legend to provide enough information to understand figures. In addition, the statistical analysis should be added to the panel to determine which differences are statistically significant.
2. Figure 3: The methylation level of CCR5 gene has been already analyzed by Gornalusse et al .. The authors should compare their primers with those designed previously by Gornalusse et al. because the authors tried to develop techniques to assess the methylation level of CCR5 gene.
3. Figures 4: The result suggests that in ART-treated HIV-1-infected individuals, the methylation of CCR5 and CXCR4 genes is suppressed. How about other genes, e.g. the HIV -1 receptor CD4 and a house-keeping gens GAPDH? Is the methylation of CCR5 and CXCR4 genes specifically reduced in ART-treated patients compared to other genes?
4. Figures 5 and 6: The authors should add the data obtained from health people because the authors mentioned the comparisons between HIV-1 patients and non-HIV-1-infected people in Lines 330-333.
5. Figures 4, 5, and 6: The results show some variations in methylation levels. Does the variation correlate with viral load? This analysis would be valuable for the HIV-1 virology field.
6. Lines 45-46: HIV-1 uses CCR5 as a main coreceptor but not CXCR4. In addition, the references 8, 10, 11 do not support the description.
7. Lines 92-94: The reference 35 does not support the description.
8. Lines 367-368: The authors mentioned “In our data”. Which data suggest/indicate that “specific CpGs were more accessible to transcription factors”? The authors should provide the data or mention which data support the description clearly.
Minor comments
1. Lines 65-66: Is there any evidence supporting “High CCR5 levels could activate resting T-65 cells”? If not, this statement should be toned down.
